# Thrust Vectoring Control for Heavy UAVs, Employing a Redundant Communication System

**DOI:** 10.3390/s23125561

**Published:** 2023-06-14

**Authors:** Mohammad Sadeq Ale Isaac, Ahmed Refaat Ragab, Marco Andrés Luna, Mohammad Mehdi Ale Eshagh Khoeini, Pascual Campoy

**Affiliations:** 1Computer Vision and Aerial Robotics Group, Centre for Automation and Robotics (CAR), Universidad Politécnica de Madrid (UPM-CSIC), 28006 Madrid, Spain; mo.aleisaackhoueini@upm.es (M.S.A.I.); marco.lunaa@alumnos.upm.es (M.A.L.); 2Wake Engineering Company, 28906 Getafe, Spain; 3Department of Network, Faculty of Information Systems and Computer Science, October 6 University, Giza 12511, Egypt; ahmed.refaat.csis@o6u.edu.eg; 4Department of Electrical Engineering, University Carlos III of Madrid, 28919 Leganés, Spain; 5Drone-Hopper Company, 28919 Leganés, Spain; 6Ahyres Company, 28905 Getafe, Spain; 7Department of Business Administration, Istanbul Aydin University, Istanbul 34295, Turkey; mehdy.al@gmail.com

**Keywords:** sliding mode, thrust vectoring control, UAV, sommunication

## Abstract

Recently, various research studies have been developed to address communication sensors for Unmanned Aerial Systems (UASs). In particular, when pondering control difficulties, communication is a crucial component. To this end, strengthening a control algorithm with redundant linking sensors ensures the overall system works accurately, even if some components fail. This paper proposes a novel approach to integrate several sensors and actuators for a heavy Unmanned Aerial Vehicle (UAV). Additionally, a cutting-edge Robust Thrust Vectoring Control (RTVC) technique is designed to control various communicative modules during a flying mission and converge the attitude system to stability. The results of the study demonstrate that even though RTVC is not frequently utilized, it works as well as cascade PID controllers, particularly for multi-rotors with mounted flaps, and could be perfectly functional in UAVs powered by thermal engines to increase the autonomy since the propellers cannot be used as controller surfaces.

## 1. Introduction

If you are far away from your enemy, make him believe that you are near. This was written by Sun Tzu, about 2500 years ago [1]. It seems that this phrase was a precursor to the subject of unmanned aerial vehicles (UAV), a technical system combining several layers to make a flying platform, from communication to control and structural systems. UAVs are almost known as aircraft without a pilot onboard, and they have become increasingly dominant due to their wide usage as remote-controlled vehicles in different fields, such as the military, firefighting, logistics, and agriculture. The abbreviation has been changed to become unmanned aerial system (UAS), to show that such a system does not only depends upon the aircraft itself, but it depends upon several important issues such as the ground control station (GCS), Communication systems with complexity, and the computing system. UAVs have several classifications based on flight endurance, weight, flying application, altitude, flight range, and the structural type [2], which, amongst the weight and endurance, are sub-objectives of this research. According to the European Union aviation safety agency (EASA) regulations for civil UASs open category, four general types are considered regarding the weight classification: Class A1 (less than 900 g), Class A2 (less than 4 kg), Class A3 (less than 25 kg), and for weights greater than 25 kg, other categories are observed that are based on some operational restrictions. Both piloting requirements and flying zones are considered [3]. In counterpart, according to the AP-3.3.7 mission qualifications of the global NATO-STANAG 4670 UAS category, they are divided into nanoscale (less than 250 g), microscale (less than 2 kg), small scale (less than 25 kg), medium scale (less than 150 kg), and large scales (more than 150 kg) [2,4]. Among the latter classification, the heavier the UAV, the more cargo payload benefits from a longer range in a single load. The medium scales are limited to a lower flight endurance and payload carriage. Therefore, many recent contexts are concentrated on heavy-lifter UAVs to improve the control and communication system for such platforms. Likewise, this research investigates a large-scale UAV of 200 kg weight, which contains several communication and control layers to maintain the safety of the flight. Meanwhile, the payload system is a releasable low-density liquid that complicates the system’s dynamic.

Heavy UAVs consist of many components, in which the most crucial ones function as a communication system that is not only in charge of internal commands transmission but also external UAVs, which can be seen as ad hoc nodes, concatenating in a subcategory of the ad hoc network called the flying ad hoc network (FANET) [4]. Several novel technologies are proposed: infrastructure-based network (IBN), wireless sensor network (WSN), wireless mesh network (WMN), and flying ad hoc network (FANET) [5,6]. Of these, cellular assistant UAV communication is a novel technology handled by multi-aerial nodes [2,7,8]. The objective of this technology is to utilize the maximum 5G and beyond network quality supported by air-to-air and ground-to-air access points to maintain robustness in the presence of disturbance. Behjati et al. [9] represented several machine learning-based visual line of sight (VLoS) models to estimate the reference signal power and quality according to various mathematical methods. They found that the quality depends heavily upon the distance of the UAV and the GCS, and the flight altitude which leads to the elevation angle. The nonlinear models conquer the linear ones due to their highly accurate prediction. Meanwhile, the authors of [10] proposed a novel MANET protocol called UAV-to-UAV (U2U) plus UAV-to-Infrastructure (U2I) communication and outlined its benefits equipped with WSN and a linear sensor network (LSN) as data collectors, regarding the latency. They implemented a dynamic system to change the communication layers via relays, considering various ratios based on the strength of every node to be exchanged; then compared the average delay, service time, and delivery ratios of several topologies. They claimed the LSNs link to each other with minimum delay when the packets are transmitted in a queue, ordered by a lesser generation time; however, their work did not include any piratical scenarios.

In this research, a newly released U2I communication system is integrated into a WSN topology and is easily installed but responds with significantly less latency, even when transmitting high-quality videos to the ground control station (GCS). In this submission, the UAV’s weight and payload type saliently impact the communications utilized, so that in a mini UAV, the communication links are significantly limited, while the danger is also negligible. In contrast, in a large case, communication is critical and if the payload material is a liquid, even more special sensors, such as thermal and chemical pressure sensors, will be needed.

On the other hand, the control system provides several challenges related to stabilizing heavy UAVs, especially when conventional solutions are not effective. Nowadays, electrical UAVs mostly work accurately and have been improved a dozen times; however, they suffer from low flight endurance, and if electrical motors are substituted by thermal engines, the controller surfaces would be changed due to the limits of thermal systems. The authors have previously discussed the limitations and solutions proposed in [11,12]. In this paper, a novel approach is put forth that, despite its rarity in the history of aeronautics, if constructed properly, may stabilize an unmanned aerial vehicle (UAV) for a long period, even in the presence of wind disturbances. More literature reviews could be found in authors’ previous works [11,12,13,14,15,16]. Briefly, the control strategy presented in this paper—thrust vectoring using flap vanes—offers various advantages over other strategies, including simplicity of servo installation and reduction in the mechanical complexities compared to those of collective pitch propellers because they have fewer movable components; more dynamic stability in attitude control, rather than rotatable hinges or ducts; more efficiency in lift generation based on flap design, as opposed to collective pitch props, which also offer faster response times, and when adjusting the flow direction and magnitude, allows for rapid changes in thrust vectoring, which can be advantageous for applications requiring agile flight control [17]; and finally, properly designed flap vanes can contribute to noise reduction. By controlling the flow patterns and reducing turbulence, flap vanes can help mitigate noise emissions, making them suitable for applications where noise reduction is a critical factor [18].

This paper is organized into five sections, as follows; Section Two discusses UAV communication subsystems; Section Three denotes the dynamic model and control; Section Four compares results, and finally, Section Five concludes the paper.

## 2. UAV Communication Subsystems

In particular, this paper represents a complete UAV system consisting of various components controlled by the autopilot (AP). The majority of connections are direct and in minor sub-components such as the cameras and lasers, an onboard computer processes the image data and collaborates with the AP in a lower level, as shown in Figure 1. This is elaborated further in Figure 2. Meanwhile, the positioning data are enhanced by the global navigation satellite system (GNSS). Moreover, the power management Unit (PMU) supplies the energy for all subsystems, which not only regulates the thermal energy to two main outputs of 12 V and 24 V but also feeds the power system. This includes the engine control unit (ECU), three internal combustion motors, and an engine monitor to demonstrate and regulate the power system in case of danger, as shown in Figure 1. Furthermore, the UAV is empowered with a redundant radio system. Principally, the command and control radio leads all the communication levels regarding internal subsystems and outperforms the U2I communication, which is further described in Figure 3. Meanwhile, in case of no functionality, the backup radio compensates the essential subsystems to follow up the last waypoints stored in the buffer to return to land in safe mode. The UAV benefits from various advantages of such redundancy, including more reliability and fault tolerance when the principal radio fails, and the second one transmits only critical telemetry to the GCS and vice versa. Additionally, when the primary communication channel experiences problems, the redundant system helps isolate and identify the source of the issue, which facilitates timely maintenance and repairs, reducing downtime and improving overall system availability. In particular, the safety package consists of a flight termination system (FTS) used to manage the whole UAV system in an emergency case, which is simplified to an electrical board that receives and stores the last flight mode and important logs with high frequency. The designed UAV in this paper has a redundant power system, comprising thermal and electrical thrusters. Electric ducted fans (EDFs) that are much smaller only maintain stability during an emergency landing and are inactive when conditions are normal so as not to impact the inlet air stream during the flight, as shown in Figure 3. In emergency circumstances, the FTS automatically activates between six to eleven seconds (to be chosen by the GCS pilots) after the PMU output power faces a sudden decrement and transfers the supply power to batteries located next to the thermal engines, which are adequate for an agile landing. Further, the system is detailed in several subsections, the AP and digital system, the power system, the safety system, and the communication package.

### 2.1. Autopilot (AP) and Digital Systems

The AP configuration system is made up of several layers that receive the actual state traumas of each component equipped with a real-time operating system (RTOS), which analyzes the data in hierarchical series and prohibits the non-critical processes from interfering with the principal functions and performing adequate safety. Moreover, the powerful NXP-based microcontroller board is equipped with a double CPU configuration to parallelize the data logging and calculation process. Meanwhile, the high-level reference data are outputted by the guidance loop that includes a flight navigation system corrected by several references, then, the low-level orientations and estimations are provided by the attitude and heading reference system (AHRS). This imports the position desired values and desired attitude angles and generates the necessary moments for the dynamics system of the UAV, which are later saturated for the actuators. Meanwhile, the controller SW loops are divided into two modes. The guidance corrections are constantly impacting the input reference values in a closed-loop system to regulate the desired outputs. In addition, the digital system contains a Jetson Xavier (https://developer.nvidia.com/embedded/learn/get-started-jetson-xavier-nx-devkit, accessed on 5 February 2023) onboard flight computer to control and analyze the camera output and stream the video in the multi-cast mode for ground observers. Meanwhile, the flight computer is connected directly to the AP, and some of the less important commands, such as navigation enhancement achieved by processing an Extended Kalman Filter (EKF) algorithm, utilizing an additional IMU to compare the data with the AP IMU and recording auxiliary telemetry.

### 2.2. Communication System

As mentioned in the introduction, the UAV is empowered by a duplicate communication system in which the principal command and control radio leads the critical commands between the autopilot and other subsystems. Explicitly, two types of communication are considered: an internal communication system and external communication with the GCS (U2I). The internal commands are transmitted by a standard RS-485 serial interface facilitated through an internal switch to receive/send the data with the least latency (less than 5 ms) and to secure the communication system. All the auxiliary connections—namely, video streaming, the onboard computer commands, and the lights—are transmitted through a separate line. Regarding the U2I communication, two antennas are installed on the UAS, as shown in Figure 3, once an omnidirectional antenna plate is utilized for distances lesser than 10 km, which is highly powerful for flights bounded in small areas, but lacks the performance to handle all the data for longer distances. Then, a yagi antenna is installed for distances longer than 10 km.

## 3. Dynamic System and Control

In order to ensure successful missions for heavy UAVs, the performance of various components and dynamic systems necessitates optimization of the overall reaction to commands, in which minimizing error time is one of the most crucial optimization criteria, i.e., reaching zero error in the higher controller loop and all of its derivatives as quickly as possible. Among the various possibilities, the vertical thermal thrusters regulate the altitude with the least error, benefiting a long flight; however, regarding the horizontal flight, the system suffers in terms of efficiency due to the latent response of motors when the speed controller unit (ECU) commands distinct spin rates in small time intervals. Therefore, one of the most efficient solutions presented is thrust vectoring control (TVC) [19], which aims for both the optimal time and the least error. Typically, TVC consists of sensors and actuators. The sensors provide information on the UAV’s attitude and motion, while the actuators adjust the flaps’ deflection angle in response to the control signals generated by the autopilot. In particular, in the case of a multi-rotor, the system could be thought of as a multi-ducted-fan (MDF), in which each duct contains a set of lateral and longitudinal flaps corresponding to a servo (the number of flap vanes could vary depending on the design, but the less servos control employed, the less functional issues and AP limitations). In this section, two studies are conducted, focusing on aerodynamic analysis and dynamic stability.

### 3.1. Aerodynamics Analysis

Optimizing the structure according to aerodynamics principles, TVC performance mostly depends on duct design and outlet duct section; the larger diameter of the outlet section, the greater the lift production and the lower the energy consumption [19], as shown in the form of continuity in Equation (Equation 1).
(1)$$\begin{array}{c} \rho A_{r}v_{r}=\rho A_{e}v_{e}\to v_{e}=\frac{A_{r}v_{r}}{A_{e}}\\ S_{r,e}=\frac{1}{4}\pi \left(D_{r,e}^{2}-D_{s}^{2}\right)\to \beta_{d}=\frac{S_{e}}{S_{r}} \end{array}$$
where, ${\left\{\right\}}_{r,e}$ refers to the rotor and exhaust prefixes, respectively, *A* is the section area, *S* is the air inlet area, which is extracted from the central spinner area, *D* is the section diameter, *v* is the airflow velocity, and $\beta_{d}$ is duct sectional efficiency. This is experimented with by expanding the exhaust duct area up to 1.7 times. The efficiency is advanced up to 1.3 times [19]. This phenomenon is also observable with flap presence, while the thruster flap vanes intrigue a product drag force that decreases the total thrust.

In addition, the blade profile leads the inlet air impacts directly on the thrust and the power consumption [20]. As shown in Figure 4, during an aerodynamic analysis with a spinning velocity up to 5000 RPM, it was proven that the higher the pitch angle at the blade tip, the more thrust is produced and more power is consumed, referring to the table in Figure 4. Likewise, the pressure drop cowling is doubled when a tip angle is implied. Additionally, the propeller after pressure (prop. aft. pressure) is merely higher with the tip angle, which demonstrates an overall better performance with the tip angle.

Furthermore, to have a better thrust, the number of blades has an outstanding effect. To investigate the importance of this and find a suitable blade number, four different types are studied, with 5, 8, 16, and 32 blades, as shown in Figure 5 and Figure 6. during the test, the spinning velocity grew up to 6500 RPM, in which, variables of thrust cowling, pressure consumption, pressure drop cowling, outlet pressure, duct outlet pressure, and the prop. aft. pressure were observed. As shown in the four plots in Figure 5, the more blades installed, the more thrust and power consumption are obtained, while the duct outlet pressure is approximately equal for the propeller with 8 and 32 blades, the two others are lesser which means also a lesser efficiency. This also could be observed in the prop. aft. pressure, which is way higher in the propeller with 8 blades. Additionally, as is shown in the middle table in Figure 5, the thrust cowling arises as the blade number increases, but also accompanies an increment in the power consumption. Therefore, comparing all increments of thrust, power consumption, and pressure drop, in which the first one is desirable and two letters are unwanted, a configuration with eight blades has the best performance that also benefits from the continuous stream at the duct exit, as shown in Figure 5. To understand the numerical data shown in Figure 4 and Figure 5, they are clarified in distinct plots shown in Figure 6.

As shown in Figure 6, to observe the best performance among four different types of propellers, four main elements are considered: power, pressure drop, thrust, and the power-to-weight ratio. The power required by the propellers is directly proportional to their thrust, so more power results in higher performance. Conversely, decreasing power reduces the propeller’s thrust output. Likewise, increasing thrust is desirable as it enables the multi-ducted fan to lift heavier loads. Then, the power-to-weight ratio represents the amount of power generated by the propeller relative to its weight, in which a higher one indicates greater efficiency and performance. Finally, pressure drop refers to the decrease in air pressure across the propeller as it generates thrust, and a moderate pressure drop is desirable for efficient operation, while an excessive one can indicate an inefficient design. In summary, the only plot that demonstrates a lesser decrement in the pressure drop, although the higher increment in the thrust is the one belonging to the propeller with eight blades. Additionally, in cases of generated thrust, 8 blades and 32 blades functioned better, suggesting that the eight-blade model is best.

According to several experiments [21,22,23], to achieve more benefits of flap vanes and less production drag, the CoM of the UAV is considered above the rotor section, i.e., the more distance between flap vanes, the more lift forces generated. Therefore, the aerodynamic forces generated by the vanes are described as follows,
(2)$$\begin{array}{c} Lift_{f}=\frac{1}{2}\rho v^{2}C_{l}A_{f}\\ Drag_{f}=\frac{1}{2}\rho v^{2}C_{d}A_{f} \end{array}$$
where $C_{l}$ and Cd are the lift and drag coefficients of the flap vane, and $A_{f}$ is the effective area toward the aerodynamic forces.

Furthermore, several clues are considered for the flap vanes, since all the vanes are installed at the duct outlet, as shown in Figure 7. Importantly, the longer duct, the higher velocity at the exit; however, it changes the CoM downward, which leads to a reduction in the dynamic stability of the whole system. Therefore, weighting these two factors, a medium height is chosen for the ducts based on experimental analysis. Generally, there are no universal geometry rules for the height of the flap vanes compared to the duct height. This depends on several elements, including the desired thrust vectoring capability, control authority, and aerodynamic performance. Therefore, the height of the flaps or vanes could be determined through aerodynamic analysis, computational fluid dynamics (CFD) simulations, or empirical testing [24]. The objective is to achieve the desired flow control and vectoring characteristics while minimizing flow separation, drag, vane-to-duct proximity, and noise production. In this paper, the height of the flaps is considered as 70% of the ducts to maintain the effectively interact with the flow passing through the duct, improving the lift force generation and controlling the overpassing of the drag forces. Meanwhile, the geometry of the flaps is considered NACA-0015 and their deflection angle is limited to $15^{\circ }$, referring to a previous experiment [12].

### 3.2. Control Strategy

A brief equation set of a robust sliding mode TVC is presented in this section for UAV multirotor systems. Generally, hexagonal cases are equipped with more motors and propellers that provide more redundancy and improve control in the case of a motor failure. Thus, hexacopters are thus a well-liked option for heavy-lift applications [25,26,27]. As mentioned in the introduction, the solution provided in this paper involves installing flap vanes at the engine exhaust, which leads to employing a ducted-fan application. Considering mentioned clues, and the position reference objective for the system, the system’s primary state matrix contains x,y,z,ϕ,θ,ψ, and their derivatives. Addressing the servos connected to the flap vanes, the whole attitude controller is upon their movements, so if every duct has four flaps at the exit, then four deflection angles per duct will be added to the system’s states, which complicates the process. Simplifying the problem, all the desired movements and turns could be summarized into the combination of all ducts, therefore, two flap vanes per duct will be sufficient for easiness in both mechanical and computational processes, which concludes 2 extra angles per duct and overall yields 12 deflection angles as additional states, $\delta_{11},\delta_{12},\delta_{21},\delta_{22},\delta_{31},\delta_{32},\delta_{41},\delta_{42},\delta_{51},\delta_{61},\delta_{61},\delta_{62}$ that in $\delta_{ij}$, *i* represents the duct number and j∈1,2 demonstrates the latitudinal or longitudinal mode of the vanes, respectively. Thus, knowing the actual and desired states of the drone, two principal modes are considered: a vertical and attitude controller, which leads to the planar controller. Among several sliding modes (SMs) design approaches, including pole placement, Lyapunov-based, optimal control-based, and model predictive methods, we chose a Lyapunov candidate that converges to zero and in a finite time and maintains there. To this end, a suitable sliding surface is defined, containing all attitude variables, which yields;
(3)$$S=\frac{sin\left(\psi_{d}-\psi \right)cos\left(\theta_{d}-\theta \right)cos\left(\phi_{d}-\phi \right)-cos\left(\psi_{d}-\psi \right)sin\left(\theta_{d}-\theta \right)}{cos(\delta_{f,d}-\delta_{f})}$$
where, ϕ,θ,ψ are roll, pitch, and yaw angles, respectively, $\delta_{f}$ is the flap vane’s deflection angle, and the ${\left\{\right\}}_{d}$ is the desired value. Conditioning the Lyapunov candidate to be positive–definite, radially bounded, and decreasing along the sliding surface, the function and its derivative of V(s) with respect to time are as follows,
(4)$$\begin{array}{c} V\left(s\right)=\frac{1}{2}S^{2}PS\\ \to dV\left(s\right)/dt=\frac{1}{2}P\left(2SdS/dtS+S^{2}dS/dt\right) \end{array}$$
where *P* is a positive–definite matrix, and dV(s)/dt must be always negative to ensure that during the sliding mode, the system is always in a neighborhood of the sliding surface and will converge to the desired state despite any disturbances or uncertainties in the system. SMC law is defined as $dS/dt=u=-k_{1}Sign\left(S\right)+k_{2}\tanh\left(k_{3}S\right)$, in which $k_{i},i\in 1,2,3$ are constants that determine the rate of convergence to the sliding surface and are obtained through the controller design process. Thus, based on the sliding surface (*S*),
(5)$$\begin{array}{c} \;dV\left(s\right)/dt=\frac{1}{2}P\left(2S\left(-k_{1}Sign\left(S\right)+k_{2}\tanh\left(k_{3}S\right)\right)S+S^{2}\left(-k_{1}Sign\left(S\right)+k_{2}\tanh\left(k_{3}S\right)\right)\right)\\ \to dV\left(s\right)/dt=PS^{2}(k_{2}\tanh\left(k_{3}S\right)-k_{1}\left|S|\right) \end{array}$$
where dV(s)/dt will be negative whenever $k_{2}\tanh\left(k_{3}S\right)<k_{1}\left|S\right|$, since $\tanh(k_{3}S)$ is bounded in (−1,1), and *P* and the constants k1,k2,k3 are all positive. This means that dV(s)/dt is negative for all non-zero values of *S* when $k_{1}>k_{2}$ and $k_{3}<1$. Therefore, it is proven that dV(s)/dt is always negative and the system converges to a stable equilibrium point, and the control law given by $u=-k_{1}Sign\left(S\right)-k_{2}\tanh\left(k_{3}S\right)$ guarantees the stability of the closed-loop system. Finally, substituting the earlier expression of *S*,
(6)$$\begin{array}{c} dV\left(s\right)/dt={\left[\left(S\left(e_{\psi }\right)C\left(e_{\theta }\right)C\left(e_{\phi }\right)-C\left(e_{\psi }\right)S\left(e_{\theta }\right)\right)/C\left(e_{\delta_{f}}\right)\right]}^{2}*\\ \frac{P[C\left(\delta_{f,d}\right)C\left(\phi_{d}\right)C\left(\delta_{f}\right)C\left(\phi \right)+S\left(\delta_{f,d}\right)S\left(\phi_{d}\right)S\left(\delta_{f}\right)S\left(\phi \right)+C\left(\delta_{f,d}\right)S\left(\theta_{d}\right)C\left(\delta_{f}\right)S\left(\theta \right)+S\left(\delta_{f,d}\right)C\left(\theta_{d}\right)S\left(\delta_{f}\right)C\left(\theta \right)]}{S\left(e_{\psi }\right)C\left(e_{\theta }\right)C\left(e_{\phi }\right)C\left(\delta_{f,d}\right)C\left(\delta_{f}\right)C\left(\phi \right)+S\left(\delta_{f,d}\right)S\left(\phi_{d}\right)S\left(\delta_{f}\right)S\left(\phi \right)+C\left(\delta_{f,d}\right)S\left(\theta_{d}\right)C\left(\delta_{f}\right)S\left(\theta \right)} \end{array}$$
where S()≅sin(), C()≅cos(), and *e* refer to the difference between the desired and actual value. Hence, considering the control law, the attitude controller with respect to the flaps vanes’ deflection angle could be rewritten as
(7)$$\begin{array}{c} U_{roll}=I_{x}\left({\ddot{\delta }}_{f,d}-k_{1}{\dot{e}}_{\phi }\right)-k_{1}\tanh\left(k_{3}S\right)\\ U_{pitch}=I_{y}\left({\ddot{\delta }}_{f,d}-k_{2}{\dot{e}}_{\theta }\right)-k_{1}\tanh\left(k_{3}S\right)\\ U_{yaw}=I_{z}\left({\ddot{\delta }}_{f,d}-k_{3}{\dot{e}}_{\psi }\right)-k_{2}\tanh\left(k_{1}S\right) \end{array}$$
where the constants $k_{i},i\in 1,2,3$ will be obtained via trial and error during the implementation, and *I* demonstrates the inertial moment. The Equation (Equation 7) states the attitude controller that leads to the position controller.

## 4. Results

Affirming the TVC algorithm proposed through the control design section, a complete platform is modeled in SolidWorks (https://www.solidworks.com/, accessed on 21 January 2023) and exported to the Gazebo (https://gazebosim.org/, accessed on 5 February 2023) dynamic environment to observe the results, in which several platforms and trajectories were tested to optimize and tune the controller gains. Meanwhile, since the control objective in this research is to stabilize a smooth movement without harsh maneuvers, the best trajectories were ones with a few sharp turns. However, these trajectories are still too long to observe the thermal engines’ performance. Hereupon, two routes are suggested in a 300×300 m^2^ area, once a circular route and then a rectangular one to be compared thereafter, where both plans are smoothed in corners. Meanwhile, in the first two plots, a realistic simulation is performed to observe the performance of the controller in the long run. The wind noise equation applied to the system is assumed to be a zero mean with a normal distribution (Gaussian), and a differential variance to be integrated with the controller input matrix, as shown in Equation (Equation 8),
(8)$$\begin{array}{c} U_{t}=U_{c}+N_{t}\\ N_{t}=\int_{-\pi }^{\pi }v\left(cos\left(\theta \right)+sin\left(\theta \right)\right)\;d\theta \end{array}$$
where the total controller input matrix ($U_{t}$) can be determined by the primary controller input ($U_{c}$) and the noise function ($N_{t}$) based on the constant velocity and the pitch angle of the drone.

In Figure 8, a semicircular route is projected in the horizontal plane, where the desired values are highlighted with the red line, and the actual UAV movements are in blue. The flight plan constantly maintains a 5 m altitude, so the vertical *Z* axis is not considered. Starting from the waypoint (0,0), the heavy UAV moves with a constant velocity of 5 m/s. It moves smoothly along the reference trajectory, containing a random noise applied to the controller to examine the performance in the long run as a steady state mode.

Likewise, as shown in Figure 9, a rectangular trajectory with more direct and longer routes was planned in the presence of a random wind disturbance, and the controller performed better because of lesser radial lines through the trajectory. In both cases, the steady-state error was less than 4%, which proves the efficiency of the SMC presented.

Comparing these results to ones obtained by a well-tuned cascade PID controller and in a fully electrical hexacopter that uses propellers as the controller surfaces for attitude, the results are shown in Figure 10 and Figure 11, in which the overall steady-state error is less than 2%. However, in sharp points and during turns, the SMC functioned better. To be precise, the main difference between the two controllers is the use of flap vanes as the controller surfaces, as shown in Figure 8 and Figure 9, which facilitates employing the thermal engines to have a way longer flight autonomy. However, this caused various uncertainties simulated by random noises, as observed in the figures.

## 5. Conclusions and Future Work

This research addresses theoretical and practical controller platforms, offering novel solutions to the vital control issues that arise when combined with communication problems in the case of multi-UAVs for remote sensing operations. In particular, a multi-ducted fan (MDF) is designed based on several considerations, including long flight endurance and transporting heavy payloads, and to maintain more stability, a hexa-duct casing is developed. Then, utilizing the aerodynamics results, a suitable duct geometry concentrating on the exhaust area is exploited to optimize the number of blades in each propeller, which leads to better performance. The presented system is powered by thermal and redundant electrical engines, in which the thermal ones generate the required thrust during the flight, and the electrical ducted fans (EDFs) survive the MDF in case of emergency. Furthermore, in order to overcome the uncertainties of the thermal thrusters, a novel robust controller based on thrust vectoring control (TVC) is presented. The results demonstrate an acceptable performance for long-range flights when compared to a tuned cascade PID performance for an ideal case, but improving the TVC’s theory and practical application is required for the design of industrial platforms. Meanwhile, a redundant communication system is carried out to support flights with medium and long ranges. Future works will include the analysis of the sensors and the useful outcomes of the enhanced system.

## Figures and Tables

**Figure 1 sensors-23-05561-f001:**
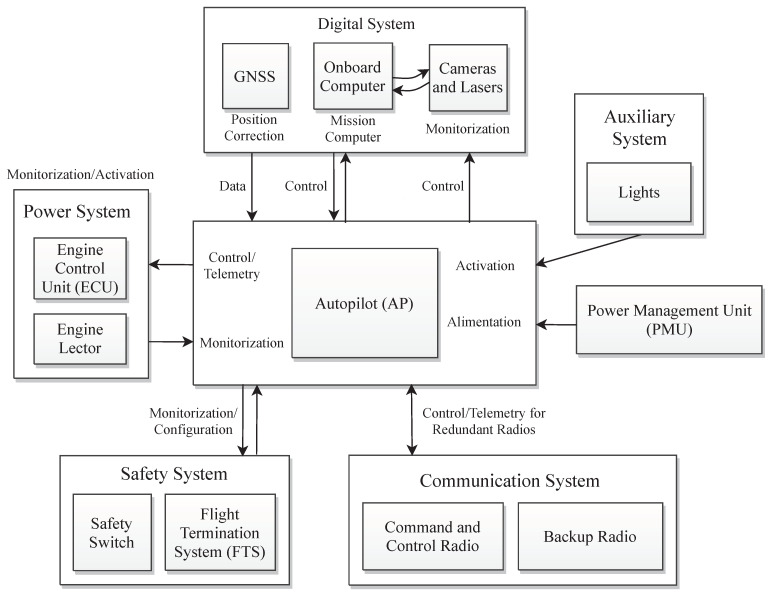
The overall schematic of the UAV system.

**Figure 2 sensors-23-05561-f002:**
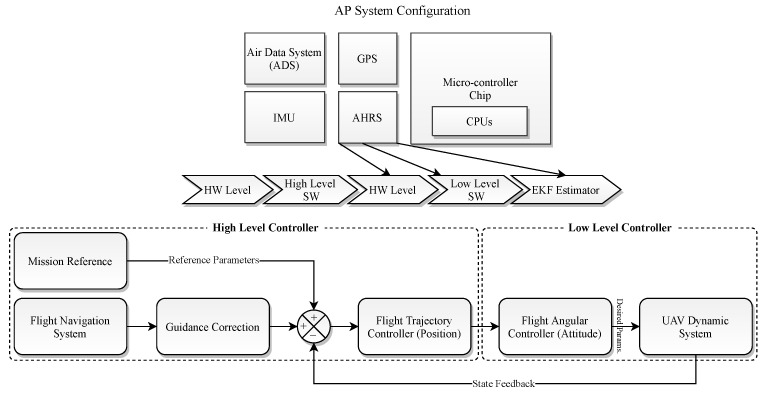
The AP infrastructure and its sub-components.

**Figure 3 sensors-23-05561-f003:**
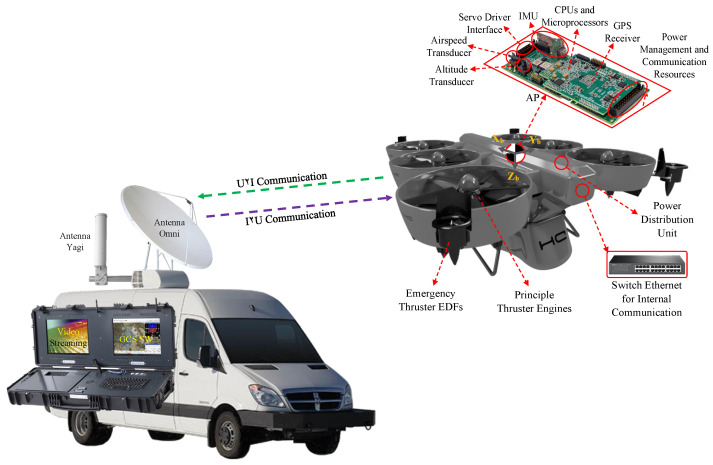
The UAS infrastructure and the communication system.

**Figure 4 sensors-23-05561-f004:**
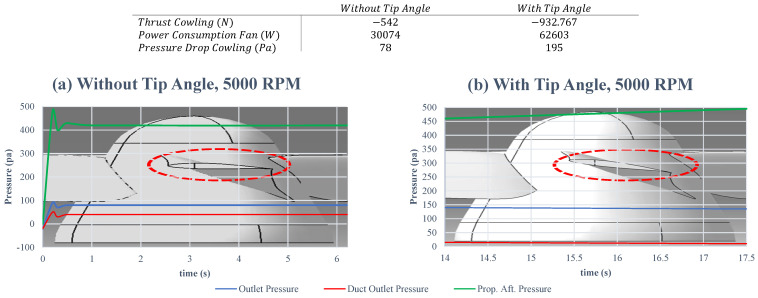
Effects of the blade tip angle on the thrust generation and power consumption.

**Figure 5 sensors-23-05561-f005:**
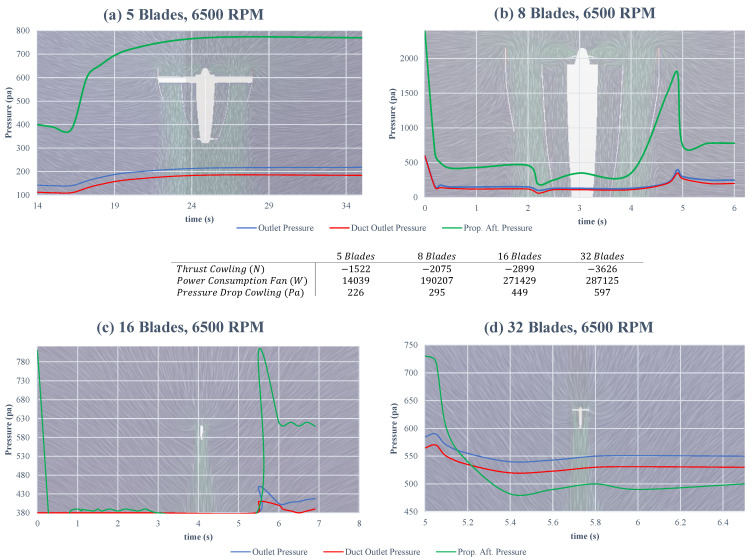
Effects of the blade numbers on the thrust generation and power consumption.

**Figure 6 sensors-23-05561-f006:**
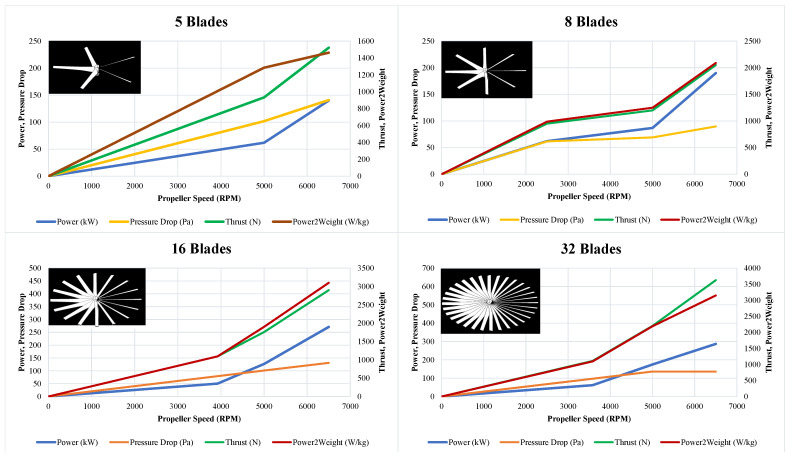
Comparison of the impact of blade number on the power, pressure drop, thrust, and power-to-weight ratio.

**Figure 7 sensors-23-05561-f007:**
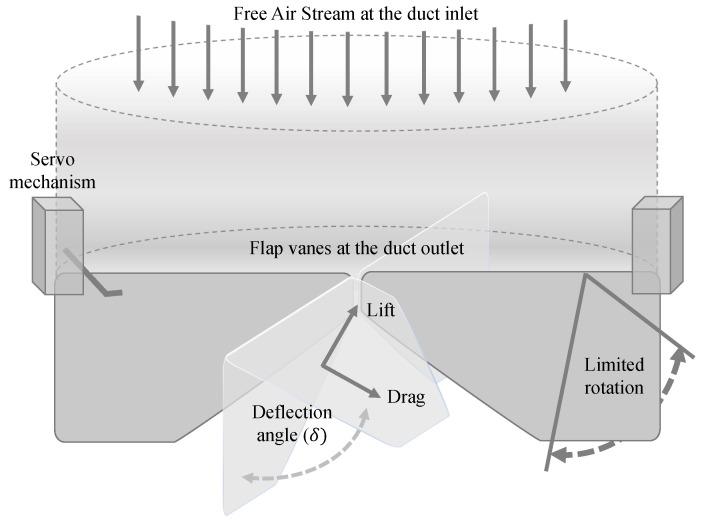
A simple schematic of the duct and the flap vanes installed at the exhaust.

**Figure 8 sensors-23-05561-f008:**
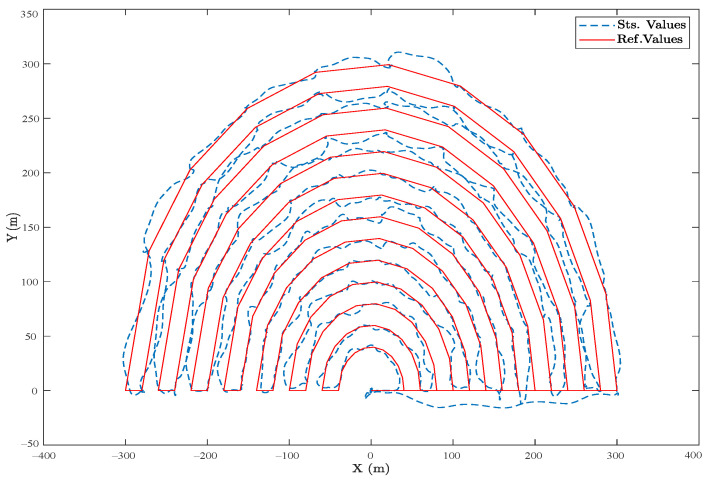
(Realistic Simulation). The horizontal projection of the reference and actual trajectory in the presence of a random wind disturbance and a semi-circular area.

**Figure 9 sensors-23-05561-f009:**
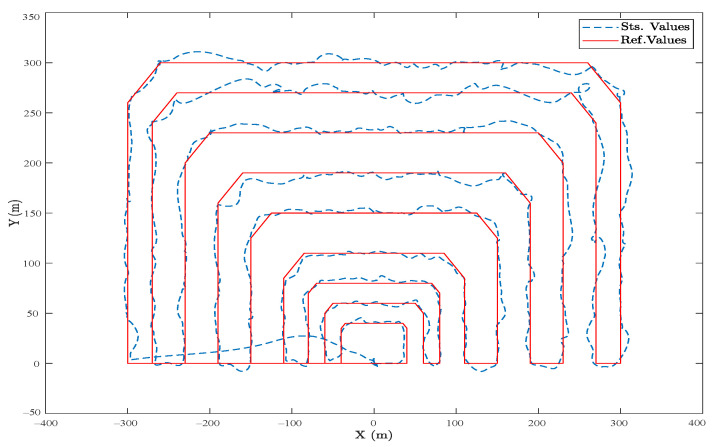
(Realistic Simulation). The horizontal projection of the reference and actual trajectory in the presence of a random wind disturbance and a rectangular area.

**Figure 10 sensors-23-05561-f010:**
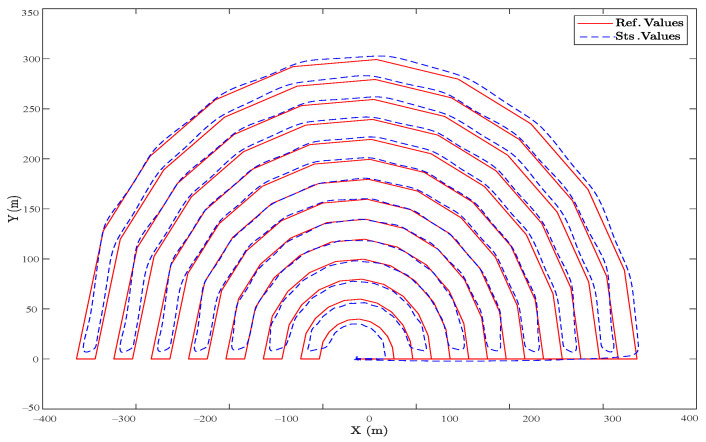
(Ideal Simulation). The horizontal projection of the reference and actual trajectory, in a semi-circular area, which is controlled by a cascade PID controller.

**Figure 11 sensors-23-05561-f011:**
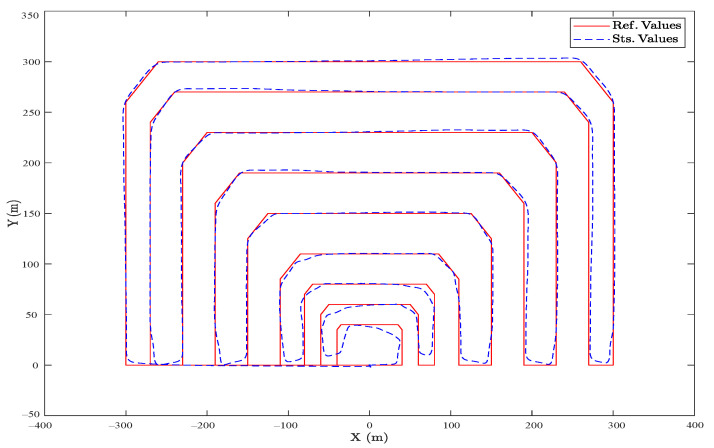
(Ideal Simulation). The horizontal projection of the reference and actual trajectory, in a rectangular area, which is controlled by a cascade PID controller.

## Data Availability

Simulation recorded videos could be found in (https://shorturl.at/uEY47) (accessed on 5 February 2023).

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
