# Peer review of "Thrust Vectoring Control for Heavy UAVs, Employing a Redundant Communication System"

_sensors, 2023, doi:10.3390/s23125561_

Round 1

Reviewer 1 Report

This article (sensors-2412649 ) examines the vector control problem of a heavy unmanned aerial vehicle. Based on the completed research work, the following aspects need to be strengthened for consideration.

1. The text mentions "multi rotor wings with flaps installed". Based on Figures 3 and 4, the thickness of the blades is small. How to install flaps in narrow spaces? Is the control of flaps reflected in the control algorithm?

2. As for "the propeller cannot be used as a controller surface", in fact, the blades have various operations such as pitch, swing, and shimmy, and the ducted rotor can also achieve complex control. What are the advantages of this system, and have the relevant mathematical descriptions been reflected?

Author Response

Thankful for your precious comments those made our work to be improved.

Reviewer 2 Report

Thank you for submitting an interesting article for review. The topic of heavy drones is currently the most up-to-date, in the light of the changing law and growing expectations in relation to drone transport systems.

Before accepting the article for publication, I would like to obtain a few clarifications regarding the presented text:
- whether and how the redundant communication system affects the presented control algorithm. I assume that since it is part of the title of the article, there should be a reference to this issue in the text. I omit the general description of redundant communication system in the second chapter,
- in the part describing the results, in Figures 6 and 7, the reference and recorded trajectories (I assume that coming from the system controlled by your algorithm) are shown. Because, as indicated in the text "a random noise applied to the controller to examine the performance" of the system, please let me know if it is possible to indicate on the same graph what was the level of noise introduced in order to be able to map how the algorithm behaved at the current moment? Otherwise, it is difficult to determine its effectiveness,
- please define the Sign(S) component appearing in expression no. 5 and in the preceding text,
- please provide information in which simulation environment the work was carried out, it is worth considering including a view from the simulation environment in the text.

To improve the readability of the text, please make the following modifications:
- the readability of Fig. No. 2 should be improved. - It is proposed to increase the size of this drawing and descriptions to the size in accordance with Fig. No. 1,
- whether the algorithm described in the article can be mapped on showed system components and properly marked in Fig. No. 2 - it would make it easier to understand the whole text, 
- please pay attention to the correct numbering of chapters - 2.0.1, 2.0.2, it should be 2.1 and 2.2,
- is it necessary to introduce the commercial name "Jetson Xavier" in chapter 2.0.1? can't you use a different form of description here in order to avoid indication of commercial component?
- the abbreviation EDF has been introduced in Figure 3 - it is not explained in the text. For the same drawing, please consider lightening the photos used to improve the readability of the drawing.
- Figures 4 and 5 contain a lot of data important for understanding the whole text, but the current form of presentation makes it practically impossible to read - please modify the way of data presention.

Minor editing of English language required.

Author Response

(The authors gave the same response as above.)

Reviewer 3 Report

This paper proposed a cutting-edge Robust Thrust Vectoring Control technique to deal with various communicative modules during a flying mission and converge the attitude system to stability. The authors explained the TVC algorithm in detail from aerodynamics analysis and control strategy two aspects. Overall, although I am not an expert in this field, I think this is a quite interesting research that has enough practice in promoting the development UAVs. As a reader of this paper, I have some questions to discuss with authors:

Firstly, as we known, for a flight termination system, appropriate redundancy devices can have certain advantages, but the degree of redundancy is worth exploring. I am curious about how this degree of degree of redundancy should be determined for a set of devices.

Secondly, in the section of Dynamic System and Control, the authors used some formulas to explore the correlation settings among various components. I noted that some hyperparameters in these formulas are set to 0.25 and 0.5. What are these settings based on?

Thirdly, to observe the thermal engines performances, a semicircular route and a rectangular route were projected to present the movement of UAV with a constant velocity. We know that there is a difference between a moving trajectory and a predetermined route. But are there any specific indicators to express these differences?

Fourthly, the conclusion section of this paper is too simplistic, and I suggest the authors supplement the statements in this section.

The English writing of this paper is acceptable.

Author Response

(The authors gave the same response as above.)

Round 2

Reviewer 1 Report

In this paper (sensors-2412649 ), the authors have made numerous modifications to showcase the details of the manuscript well.

1. In Figure 3, the drone here is equipped with many rotors and culverts. Due to the lack of good aerodynamic design for the large rotor duct shell, how to ensure smooth intake of small rotors near the large duct without interference?

2. Is the influence of turbulence at the duct lip on the small rotor neglected during pitch control? 

Author Response

we would be thankful for your precious comments that helped us to improve the work several times

Reviewer 2 Report

Thank you for making corrections to the text. The most important comments have been included in the text, appropriate corrections and extensions of descriptions have been introduced.

However, despite the fact that the vast majority of texts will be read in electronic format, the presentation of data in Figures 4 and 5 needs to be refined. As mentioned in an earlier review, the current presentation makes it difficult to perceive these very interesting results. I would consider reformatting the way these images are presented.

Author Response

Thankful for your comment.
